# What’s in a Gene? The Outstanding Diversity of *MAPT*

**DOI:** 10.3390/cells11050840

**Published:** 2022-03-01

**Authors:** Daniel Ruiz-Gabarre, Almudena Carnero-Espejo, Jesús Ávila, Vega García-Escudero

**Affiliations:** 1Anatomy, Histology and Neuroscience Department, School of Medicine, Universidad Autónoma de Madrid (UAM), 28029 Madrid, Spain; daniel.ruiz@cbm.csic.es (D.R.-G.); almudena.carnero@estudiante.uam.es (A.C.-E.); 2Centro de Biología Molecular Severo Ochoa (UAM-CSIC), 28049 Madrid, Spain; 3Graduate Program in Neuroscience, Universidad Autónoma de Madrid (UAM), 28029 Madrid, Spain; 4Networking Research Center on Neurodegenerative Diseases (CIBERNED), Instituto de Salud Carlos III, 28031 Madrid, Spain

**Keywords:** *MAPT*, Tau protein, alternative splicing, intron retention, Alzheimer’s disease

## Abstract

Tau protein is a microtubule-associated protein encoded by the *MAPT* gene that carries out a myriad of physiological functions and has been linked to certain pathologies collectively termed tauopathies, including Alzheimer’s disease, frontotemporal dementia, Huntington’s disease, progressive supranuclear palsy, etc. Alternative splicing is a physiological process by which cells generate several transcripts from one single gene and may in turn give rise to different proteins from the same gene. *MAPT* transcripts have been proven to be subjected to alternative splicing, generating six main isoforms in the central nervous system. Research throughout the years has demonstrated that the splicing landscape of the *MAPT* gene is far more complex than that, including at least exon skipping events, the use of 3′ and 5′ alternative splice sites and, as has been recently discovered, also intron retention. In addition, *MAPT* alternative splicing has been showed to be regulated spatially and developmentally, further evidencing the complexity of the gene’s splicing regulation. It is unclear what would drive the need for the existence of so many isoforms encoded by the same gene, but a wide range of functions have been ascribed to these Tau isoforms, both in physiology and pathology. In this review we offer a comprehensive up-to-date exploration of the mechanisms leading to the outstanding diversity of isoforms expressed from the *MAPT* gene and the functions in which such isoforms are involved, including their potential role in the onset and development of tauopathies such as Alzheimer’s disease.

## 1. Introduction

Tau protein belongs to the microtubule-associated proteins (MAP) family and is encoded by the single-copy microtubule-associated protein Tau gene (*MAPT*), which is located on chromosome 17q21 in humans and consists of 16 exons [1,2,3]. Tau participates on different physiological functions, including microtubule assembly and stabilisation [3], neurite outgrowth and axonal transport [3], and regulates neuronal activity, neurogenesis and long-term depression (LTD) [3,4].

However, Tau is also involved in a number of pathological processes, undergoing misfolding and oligomerisation into paired helical filaments (PHFs) and neurofibrillary tangles (NFTs) [5]. These neuropathological lesions constitute a characteristic hallmark of a wide range of tauopathies, including Alzheimer’s disease (AD), progressive supranuclear palsy, corticobasal degeneration, argyrophilic grain disease, Pick’s disease, Huntington’s disease or frontotemporal dementia with parkinsonism-17 [3,6].

The reasons behind the shift from a physiological to a pathological state are not clearly elucidated, but it is rather well-established that Tau’s post-translational modifications (PTMs) are crucial for its normal function and thus, alterations in the pattern of such modifications may be responsible for the transition from a healthy soluble protein to insoluble misfolded fibrils of Tau [5,7]. Two PTMs of Tau are mainly related to neurofibrillary degeneration: hyperphosphorylation and truncation [5], both leading to neurotoxic gain of function and generally aggregation-prone versions of Tau [1,5,6,8]. Besides hyperphosphorylation and truncation, Tau is post-translationally modified by a great deal of other processes, including ubiquitination, SUMOylation, glycation, acetylation, glycosylation, O-GlcNAcylation, and nitration—as recently reviewed by Alquezar et al. [7].

In addition, more subtle nuances in Tau function can be explained by the existence of different Tau isoforms, generated by alternative splicing [9]. Alternative splicing (AS) is a co- or post-transcriptional process that occurs when introns of a certain pre-mRNA are excised in more than one way giving rise to structurally and functionally different protein isoforms from the same gene [10,11].

AS is a powerful regulatory mechanism that affects at least 60% of human genes, and has been hypothesised to be responsible for the greater proteomic and cellular complexity of higher eukaryotic organisms [9,12,13]; although more recent proteomic approaches point out that the contribution of this mechanism to proteomic complexity may be wildly overestimated [14]. Nevertheless, it should be pointed out that certain genes, including *MAPT*, have been robustly determined to undergo alternative splicing, yielding a variety of splicing-generated isoforms [14,15].

This is particularly relevant, considering that alternative splicing is a regulated process that suffers alterations both during healthy ageing and under pathological conditions [16,17,18]; some of which are disease-specific and pointedly involve Tau [16,19,20]. 

The purpose of the present review is to highlight the great diversity of Tau isoforms mainly generated by alternative splicing and how this entails a consistent diversity function-wise; both of which get markedly and specifically modified both with natural ageing and with age-related disorders.

## 2. Alternative Splicing: A Force of Diversity

Alternative splicing is not constituted by one single mechanism acting throughout the genome generating spliced alternatives. Instead, different types of alternative splicing events have been identified, including exon skipping, mutually exclusive exons, alternative use of 5′ splice sites (5′ss) and 3′ splice sites (3′ss), alternative polyadenylation and intron retention [9,12,17] (Figure 1). 

The most common type of AS is exon skipping, also known as cassette exons [12]. Cassette exons are delimited by splice sites located at the boundaries between mRNA-coding and non-coding sequences and may be included or not in the mature mRNA transcript [21,22,23]. When the exon is retained, the splicing pattern is similar to that of a constitutive gene; however, when it is removed, it is presumably spliced out together with its flanking introns [23,24]. 

Mutually exclusive exons refer to a specific type of splicing in which only one of two or more candidate exons in a cluster is included into the mature mRNA, but the exclusion or inclusion of both simultaneously does not occur [23,25]. 

Alternative 3′ and 5′ splicing events constitute at least one-quarter of the known AS events [12]. Alternative 5′ splicing sites exons (A5Es) and alternative 3 ’splicing sites exons (A3Es) are flanked on one end by a constitutive splice site and on the other end by two (or more) alternative splice sites, resulting in different primary transcripts of the same gene [22,24].

Polyadenylation is the process whereby almost all eukaryotic mRNAs acquire an uncoded polyA tail at their 3′ ends. Alternative polyadenylation is defined as the use of more than one polyadenylation signal present in pre-mRNAs, allowing a single gene to encode a variety of mRNA transcripts [26]. 

Finally, intron retention is one of the least common types of alternative splicing, by which an intron sequence is maintained in the mature mRNA molecule due to its weak flanking splice sites [21,24,27]. 

Transcripts from the majority of human protein-coding genes may undergo one or more forms of alternative splicing that, combined, can give rise to a number of different alternative spliced isoforms. All these events require the spliceosome [28], a multi-subunit ribonucleoprotein complex, composed of five small nuclear ribonucleoproteins (U1, U2, U4, U5 and U6), acting along many other proteins (more than a hundred, by some estimates) [17]. The spliceosome recognises three cis-acting elements on pre-mRNA: the 3′ splice site, the 5′ splice site and the intronic branch-point sequence [29]. However, the regulation of the whole process is much more complex and depends not only upon these cis-acting elements, but also other *cis*-acting factors (exonic and intronic silencers and enhancers) and trans-acting factors that interact with these and include activators (such as proteins of the serine-rich family) and repressors (heterogeneous ribonucleoproteins, for instance) [17].

Thus, AS constitutes a tightly regulated process, controlled by multiple exonic and intronic cis-elements and trans-acting splicing factors [17,18,30]. The whole process, including these splicing factors, is altered in healthy ageing and age-related pathologies [16,31]; consequently affecting this very regulation and all the mechanisms in which AS may be involved, which is especially relevant for the case of the *MAPT* gene and the associated protein isoforms of Tau. 

## 3. Tau Alternative Splicing: Diversity of Forms

The human microtubule-associated protein Tau gene (*MAPT*) consists of 16 exons, with at least 6 of them being subjected to alternative splicing [3,9]. Exons 1, 4, 5, 7, 9, 11 and 13 are the constitutive exons of the Tau molecule [3]. Exons 0 (also termed as exon -1 by some authors) and a part of exon 1 encode the 5ʹ untranslated sequences of *MAPT* mRNA. Exon 1 encodes the N-terminus, while exons 4, 5, 7 and the first part of exon 9 encode the region between the N-terminal inserts and the first microtubule-binding repeat, including the proline-rich regions (mainly encoded by exons 7 and 9). Exons 9, 11 and 12 encode the three constitutive tubulin-binding repeats and exon 13 is translated into the C-terminus of Tau [3,9,32,33] (Figure 2A).

On the other hand, at least exons 2, 3, 4A, 6, 8, and 10 undergo alternative splicing. Six major Tau isoforms can be found in the human central nervous system (CNS) through different combinations of the splicing of exons 2, 3, and 10 [9,34], ranging from 48 to 67 kDa [33] (Figure 2B). Exons 2 and 3 encode two N-terminal inserts and can be included or excluded together, but only exon 2 can be included on its own, since, strikingly, exon 3 needs to be included in conjunction with exon 2 [3,35], although the mechanisms reigning this conditional inclusion of exon 3 remain unexplained. As for exon 10, it encodes the second of four possible tubulin-binding repeat regions, that appear right between those encoded by exons 9 and 11. Inclusion of exons 4A and 6 is restricted to certain tissues [9] and their inclusion generates tissue-specific isoforms (Figure 2B); while exon 8 has been described in other mammals, but not in humans [36].

Recently, a new Tau isoform named W-Tau arising from intron 12 retention has been described, opening new research avenues that focus on the generation of novel isoforms by means of less frequent forms of alternative splicing [37] (Figure 2B). Of note, *MAPT* intronic sequences have been studied in different contexts. For instance, intron 9 contains a nested cryptic exonic sequence, whose translation generates a protein named saitohin [38] (Figure 2A); suggesting that the translation of introns from *MAPT* may be exhaustively regulated.

Lastly, exon 14, although considered alternatively spliced, is part of the 3ʹ untranslated region and therefore does not imply any changes in the composition of the protein [3,9].

The regulation of these splicing events is complex and involves the orchestration of a myriad of splicing factors, including Serine-rich splicing factors (SRSF1, 2, 3, 4, 6, 7 and 9) [34,39] and the arginine and serine-rich coiled coil protein (RSRC1) [40], but also non-SR proteins, such as RNA-binding motif protein 4 (RBM4) [34] and 11 (RBM11) [40], RNA helicase p68, heterogeneous nuclear ribonucleoproteins (hnRNP) [41] or Tra2β, among others [34,39]. Such regulation is still more intricated, considering the Tau splicing depends upon developmental stage and tissue type [9], the modulation of the activity of such factors [34] and the existence of different RNA structures that can influence *MAPT* mRNA stability and splicing [42]. The complexity of regulation of *MAPT* splicing highlights the need of finely tuned mechanisms that ensure the correct and precise modulation of the processes that encompass such splicing [40,41]. 

However, it is not the aim of this review to discuss the mechanisms regulating the process of alternative splicing of the *MAPT* gene, but rather, explore the consequences of said process. Hence, isoforms arising from all the described splicing events will be described in the following sections and are summarised in Figure 2B.

It is important to mention that the majority of these splicing events are not mutually exclusive and can therefore happen simultaneously. Also, as a result of the splicing process, a shift on the reading frame or the appearance of a different termination codon can occur, thus resulting in a different protein or in the premature truncation of the protein [43,44,45]. All this together can give rise to an even greater number of isoforms. For representation purposes only, Figure 2 shows the variants of Big Tau, isoforms containing exon 6 and W-Tau containing 4 microtubule binding repetitions and 2 amino-terminal inserts, but any and all of the combinations of expressions of exons 2, 3 and 10 may be possible for these isoforms. 

Finally, given the human-specific nature of Tau-related disorders such as Alzheimer’s disease and the broad interspecies variability of the *MAPT* gene and the regulation of its alternative splicing events [46,47], we will only dwell on human Tau isoforms. Thus, we will not consider exon 8-including isoforms, which have been described in goat, rhesus monkey and bovine brains, but not in humans [36]. 

### 3.1. Central Nervous System Isoforms: Meet the Classics

Classically, researchers describe six main Tau isoforms existing as proteins in the human brain ranging from 352 to 441 amino acids and arising, as mentioned, from the inclusion or exclusion of exons 2, 3 and 10 [3,9,32].

The default splicing pattern of exon 2 is inclusion, and its pre-splicing enhances the consequent inclusion of exon 3, although this latter follows exclusion as its most common splicing pattern. In fact, as mentioned, exon 3 requires the inclusion of exon 2 to be able to be included itself, a process which mechanisms have not yet been fully explained [35]. Alternative splicing of exons 2 and 3 result in Tau isoforms with 0, 1, or 2 inserts in the N-terminal domain, known as 0N, 1N, and 2N isoforms, respectively [2,48]. 

As for exon 10, the choice between its inclusion or skipping gives rise to Tau isoforms with four (4R) or three (3R) microtubule-binding repeat domains in the C-terminal end [49]. These isoforms differ from each other in their affinity for microtubules [50], with 4R isoforms showing increased tendency to bind to microtubules and greater potency in inducing their assembly [32].

The default behaviour of exon 10 is inclusion; however, its flanking exons influence the splicing: upstream exon 9 promotes its inclusion, while downstream exon 11 competes with it [9].

The isoforms resulting from the interplay of inclusion and exclusion of exons 2, 3 and 10 are usually named after the number of 29-amino-acid N-terminal inserts encoded by exons 2 and 3, and 31-amino-acid microtubule-binding repeats they include. Namely, we can find Tau with four repetitions harbouring two inserts (4R2N), one insert (4R1N) or no insert at all (4R0N), and the same for 3R Tau (3R2N, 3R1N and 3R0N) (Figure 2B). 

Although these isoforms are usually mentioned as if they were present in equal amounts, the brain pattern of expression of these proteins is not such [32,51]. 4R and 3R Tau isoforms do seem to be found in almost an equimolar ratio in the adult human brain, but 0N, 1N and 2N isoforms are expressed very differently, with 2N constituting only around 9% of the total Tau, 1N more than half (approximately 54%) of total Tau and 0N roughly 37% of total Tau [51]. The balance between 4R and 3R isoforms have been deemed to be key for brain function, with splicing dysregulation being involved in the development of tauopathies [10]; but rather little research exists regarding 0N, 1N and 2N isoforms’ proportion.

These data hold true for adult brains, but it is important to note that alternative splicing of the *MAPT* gene is also subjected to temporal regulation, with a marked shift on the expression of exons 2 and 10 mainly during the perinatal period [52], while the variation of exon 3 expression is smaller in comparison, but still significant. Such changes imply the shift from a hyperphosphorylated-3R0N-predominant environment in the foetal brain to the landscape of isoforms described above for the adult one. This shift is preserved in every vertebrate species studied to date, although the result is not always the same: while human adult brains keep a similar level of 3R and 4R Tau isoforms, adult mice express solely 4R isoforms, while adult chicken brain contains 3R, 4R and 5R isoforms [32,52].

The functional implications of this shift are not completely elucidated, but the fact that it is an evolutionarily conserved mechanism points towards a possible role in ensuring the versatility of the protein to accomplish brain development and plasticity during prenatal stages but also microtubule stabilisation and axonal transport in the adult brain [32,52]. 

Again, researchers rarely make a distinction on the localisation of these isoforms within the cell when discussing their presence in certain tissue, but studies aimed to examine this prove that subcellular localisation seem to be isoform dependent [53]. Indeed, isoform localisation preference exists between developmental stages, tissues, cell lines, brain regions and intracellular compartments [54]. Hence, 2N isoforms are retained in the soma [33], while 1N isoforms localise in the nucleus [53,55] and 0N isoforms can be found in both somas and axons [53]. However, the difference between 4R and 3R Tau does not seem to make a difference in Tau’s axonal sorting [33]. 

Nevertheless, to our knowledge, a subcellular, compartment-specific mapping for the different Tau isoforms do not exist for human neuronal cells. This, together with the fact that many studies do not report specific isoforms when discussing Tau-related results, makes it rather difficult to ascertain a specific function or localisation to each isoform. Future research should consider this gap in the literature and explicitly report the isoform or isoforms involved in their findings.

Finally, alternative splicing of the *MAPT* gene is tissue-specific, as proven by the differential expression of certain isoforms, such as big Tau or exon 6-including isoforms [9,52,56]; but also cell-type and cell-stage specific [52]. This fact, together with the different susceptibility of brain regions to the development of tau-related pathologies, such as Alzheimer’s disease [39] or paranuclear superior palsy [57] and the region-specific nature of other tauopathy-related mechanisms, such as Tau post-translational modifications [58] underline the importance of considering also regional variability of Tau isoforms and their potential role in determining such differential vulnerability. This specific area, however, remains understudied to date and further research is needed supported by means of novel technologies [59].

### 3.2. Big Tau: A Giant Outsider

The last decade of the past century witnessed the discovery of a version of Tau protein with higher molecular weight in the dorsal root ganglion (sensory neurons) and pheochromocytoma cells of neural crest origin (PC12 cells), both from rats [60,61]. Later on, it was also found in the optic nerve and in CNS cells with projections to the PNS [56].

This high-molecular-weight Tau was termed Big Tau and was determined to arise from an 8–9 kb mRNA from the *MAPT* gene, longer than the previously described (~6 kb), due to the inclusion of another exon between exons 4 and 5, which has been named exon 4a [30,62,63]. Actually, several isoforms of Tau containing exon 4a may exist, since some of them were also found to contain exon 6 [9,63] (Figure 2A). 

However, data is lacking in human tissue regarding these Big Tau isoforms. Most of the information we have so far has been attained through genomic analysis based on transcript alignments [56]. We do know that the default pattern for this exon is exclusion and, due to its length, it is expected to require a helper to be included, encoding a 251-residue fragment that results in a great extension of the amino-terminal region [9] (Figure 2B).

### 3.3. Black Sheep: Isoforms including Exon 6

Another exon that is not present in the six main Tau isoforms from human CNS is exon 6. Many authors have repeated that exon 6 is not expressed in the brain and relegated it to peripheral tissue such as muscle [3,34], but evidence traceable back more than 20 years point that Tau isoforms expressing exon 6 can be found as a protein in foetal and adult brain [64], including forebrain, hippocampus and cerebellum; albeit it is indeed more prominent in skeletal muscle and the spinal cord [64,65]. 

The splicing behaviour of exon 6 is most frequently inclusion, although it strongly depends upon its flanking exons: upstream exon 5 promotes its inclusion, unlike downstream exon 7 that competes with it [9,66]. In addition, exon 6 has proven to include two alternative 3′ splice sites, one closer to the beginning of the exon (6p, or proximal) and the other one a bit further (6d or distal); thus yielding three possible isoforms [64] (Figure 2B). When used, these alternative sites cause a frameshift that finds a premature stop codon, giving rise to two truncated proteins that include the N-terminal region but lack the proline-rich region, the microtubule-binding domain, and the C-terminal region of canonical Tau proteins [9,65]. Such a frameshift results in the appearance of specific sequences for each one: PCCVPRATFLS for 6p isoforms and FWSKGDETQGG for those that use the distal site (Figure 3A). Importantly, these would be the only Tau isoforms to lack the microtubule-binding domain, which begs the question as to whether they can be considered Tau isoforms at all if we were to focus solely on its function, given that the very core function of Tau as a member of the microtubule-associated proteins, would be related to microtubules. Nevertheless, Tau protein has been linked to a multitude of functions during the last decades of research [67,68,69]. Additionally, there is certain degree of redundancy of these functions with other microtubule-associated proteins such as MAP2 and neurons from Tau knockout mice display an almost identical morphology but altered synaptic functions with respect to wild-type [70,71]. Together, these data may suggest that Tau protein is much more than a microtubule-associated protein, and this might not be considered to be the main function of this protein. 

Within exon 6′s default pattern of inclusion, not all of these isoforms are equally included. The regulation seems to be related to the affinity of the site to splicing machinery and 6p isoforms constitute the predominant species [9,66]. In addition, the expression of 6p and 6d isoforms is regulated spatially [72] and temporally [45], being 6d levels higher in foetal brain, while 6p isoforms are present similarly in foetal and adult brains. Within the adult brain, both 6p and 6d can be found in different CNS areas (including cortex and hippocampus), but display the highest levels in spinal cord and cerebellum [45,64]. Precisely in cerebellum, 6d isoforms’ levels were comparable to those of full-length Tau isoforms [45]. In any case, a 6d isoform-specific antibody show this isoform is not present in neurofibrillary tangles. Together, these results are especially interesting, given the cerebellum’s lack of Tau-related pathology in Alzheimer’s disease patients.

### 3.4. W-Tau: The Rara Avis

Very recently, the landscape of *MAPT* splicing variants has become even more complex, with the discovery of novel Tau isoforms generated by intron 12 retention and the consequent translation of a fragment of that intronic region [37] (Figure 2B).

Intron retention is the most common type of alternative splicing event in different organisms, including plants, fungi and unicellular eukaryotes, but it has not been until recently that its role in humans and other mammals has been noticed and begun to be regarded as a regulating mechanism for many physiological and pathological events [73], with important contributions to cellular homeostasis [74] and an age-dependent regulation that increases these events with age [75].

In this case, intron 12 retention in the *MAPT* gene implies the appearance of a premature stop codon that translates into a protein lacking Tau’s C-terminal region, but having a unique 18-amino-acid sequence right after the fourth microtubule binding repeat encoded by exon 12 [37] (Figure 3B). Interestingly, the sequence corresponding to the fragment of intron 12 that is retained is translated as KKVKGVGWVGCCPWVYGH, which contains two tryptophan residues (W), an amino acid that cannot be found at any other point within the Tau molecule; hence the name proposed by the authors: W-Tau.

Importantly, even though there is a high degree of interspecies homology for Tau protein, pointing to *MAPT* exons being conserved phylogenetically, that is not the case for introns, not even in the case of chimpanzees, which express an identical Tau molecule to that found in humans [46]. Consequently, W-Tau is human-specific, as are some tauopathies, such as Alzheimer’s disease, which is not accurately mirrored in animal models [76].

RNAseq data pointed out that mRNA for W-Tau is expressed in ~50–75% of humans, depending on the brain region examined, with frontal cortex displaying higher levels than frontal lateral cortex and hippocampus [37], newly pointing to the regionally-specific nature of *MAPT* alternative splicing. Strikingly, those results imply that not everyone expresses this isoform, at least in the regions that were analysed, or that the expression is so low that it falls below the technique’s detection threshold. The study of W-Tau mRNA and protein levels in other cerebral and peripheral areas might help understand better the processes behind its modulation. In addition, W-Tau protein levels are diminished in Alzheimer’s disease patients with respect to control, non-demented subjects, thus suggesting a role in the development and progression of the disease [37]. 

Although the mechanisms responsible for this unique splicing event are not fully elucidated, the authors found a possible inverse relationship between W-Tau levels and the GSK3β mediated activation of the splicing factor SRSF2 (also named SC-35) [37], a member of the SR-protein family, which have been previously linked to intron-modulating splicing events [77]. Noteworthily, this mechanism of splicing regulation has been also related with exon 10 inclusion [78,79]

Since the alternative splicing event that spawns W-Tau isoforms is located almost at the end of the molecule, it does not directly interact with any other of the previously mentioned splicing events, meaning they are all—at least theoretically—compatible with this one. Hence, this would greatly increase the repertoire of potential isoforms generated from alternative splicing from the 30 mentioned by Luo et al. [65] to at least 54, not considering post-translational modifications, that drive by themselves numerous modification in Tau’s structure and function [58].

## 4. Tau Alternative Splicing: Diversity of Functions

The generation of different isoforms of any protein by means of alternative splicing frequently implies different functions for those isoforms, or mechanisms of self-regulation between them. In the case of *MAPT*, the function of Tau isoforms is not completely clear, since the precise structure of each isoform is not conserved between species [32]. However, research has pointed out that certain regions of the final protein are related to specific functions, such as microtubule stabilisation and polymerisation or interaction with other proteins [3]. This, together with the fact that Tau isoforms are regulated temporally [52] and spatially [52,72,80], suggests that specific Tau isoforms carry out specific cellular functions.

The result of the above described mechanisms of alternative splicing is a number of isoforms with distinct fragments, but in terms of sequence and biochemical properties, Tau consists of four major domains: the N-terminal end (NTR), the proline-rich region (PRR), the microtubule-binding domain (MTBD) and the C-terminal end (CTR) [3,4,81] (Figure 4A). The N-terminal end is of acidic nature and its negatively charged at physiological pH, but the proline-rich region and the MTBD are markedly basic, generating somewhat of a dipolar structure on the molecule [67].

As for the extension of these areas, the N-terminal region ranges from the beginning of the molecule until the first residues of exon 7, encompassing residues 1–151 of the Tau 441 isoform. The proline-rich region is encoded by parts of exons 7 and 9, which constitute the proline-rich regions 1 and 2, respectively, while the microtubule-binding domain is made up by the rest of exon 9 and exons 10 (only in 4R isoforms), 11 and 12. Lastly, exon 13 constitutes the C-terminal end (Figure 4A). Other authors consider that the molecule of Tau can actually be divided in just two regions, attending to functional factors: the projection domain, that includes the N-terminal region and the first proline-rich region, and the microtubule-assembly domain, composed of the second proline-rich region, the microtubule-binding tandem repeats and the C-terminal end. Both classifications do not need to be mutually exclusive (Figure 4A).

Such functionality is intimately linked to alternative splicing, since splicing events suppose the extension, reduction or even the deletion of certain of these regions; or modifications in their interactions through changes on the distance between them. For instance, isoforms including exons 2 and 3 have a larger N-terminal region and those including exon 10 have one extra repetition that accounts for a 31-amino-acid longer microtubule-binding domain [32] (Figure 4B). 

Isoforms including exon 4a and exon 6 on the canonical splicing site (6+) display a longer molecule. Inclusion of exon 4a entails a great amplification of the amino-terminal region [56]; but the inclusion of exon 6, due to its composition, implies an elongation of the proline-rich region [64,65] (Figure 4C).

Finally, inclusion of exon 6 on either the proximal (6p) or distal (6d) alternative 3′ splice site or retention of intron 12 generating W-Tau isoforms is linked to the loss of the C-terminal end of the molecule (Figure 4D). In the case 6p and 6d isoforms, the loss is more dramatic, because it entails the deletion of most part of the molecule [64], leaving only the N-terminal region and a small sequence whose function is not clearly determined, corresponding to the translation of the sequence of exon 6 under the new frames. On the other hand, W-Tau isoforms lack the C-terminal region encoded by exon 13 but keep the rest of the molecule in its entirety [37] (Figure 4D).

There is a considerable gap in the literature regarding the specific function of each Tau isoform, most likely due to the difficulty of asserting a specific function of the protein to a specific isoform. However, there is some research focused on ascertaining the functions of specific fragments and that, coupled with the proportion in which each isoform is present [51] can be used as an approach to the function of isoforms carrying such fragments; albeit it is important to bear in mind that a lot of the functions of these regions overlap and are also dependent on the intramolecular interactions between them [67]. 

### 4.1. The Projection Domain: Tau’s Versatile N-Terminal End and the Extension of Big Tau

Research dealing with Tau functions has classically focused on the microtubule-binding region, but the past decades have seen an increasing interest rise towards the implications of the N-terminal end of Tau isoforms in physiological and pathological conditions.

The so-called projection domain receives its name because it projects away from the microtubule surface when the microtubule-binding region is attached to these. Such position grants this region the opportunity to interact with other cytoskeletal and cytoplasmic proteins.

Within the projection domain, the N-terminal end constitutes the least evolutionary conserved region of Tau [67], with the sole exception of both amino-terminal inserts encoded by exons 2 and 3, which points to efficient interactions with specific ligands [9], probably annexins [67]. Indeed, the whole region has proven to interact with a myriad of cytoplasmatic membrane proteins beyond annexins [9,82], including synaptic vesicle-associated proteins, such as synapsin-1, synaptogyrin-3 and synaptotagmin-1 [67]. These interactions could be related to recent evidence that suggests that pathological cleavage of the NTR contributes to early synaptic failure in Alzheimer’s disease [83] and that the mutation A152T within this region prompts presynaptic dysfunction [84]. Analogously, this region is able to interact with other membranous elements in the cell, such as mitochondria or other organelle’s membranes [81].

The majority of these NTR–membrane interactions are susceptible of regulation via (de)phosphorylation [67,85], which suggests the existence of regulatory mechanisms via intracellular signalling. In fact, membrane-associated Tau is dephosphorylated at serine and threonine residues [69]. This may help explain the contribution of Tau’s hyperphosphorylation to the pathology of Alzheimer’s disease since, in addition to promote Tau self-aggregation, it may hinder other membrane-related functions.

In line with this, it should come as not much of a surprise, then, that this region interacts with several signalling and phosphorylation-related proteins, such as GSK3 or different members of the 14-3-3 proteins family [67], although the latter also interact with other Tau regions, so it might not display a specific interaction with this area. 

The disruption of the membrane-related functions of Tau’s N-terminal region could have direct implications in pathological conditions, contributing to Tau aggregation and toxicity, and to the localisation of Tau back from the axon to the soma [67]. 

Other of the most studied functions of Tau is axonal transport [3]. The NTR has been proposed to be at least partly responsible of accomplishing this function, since it directly binds to the C-terminal area of the p150 subunit of the dynactin complex [86].

As mentioned before, Tau subcellular localisation is isoform-dependent, with 1N isoforms specifically being targeted towards the nucleus [53]. This suggests a functional relation between exon 2 and nuclear Tau, but only when it does not appear accompanied by exon 3 [55]. Strikingly, there is no nuclear localisation signal within the sequence of Tau corresponding to exon 2 that explains why 1N isoforms may be directed toward the nucleus, so it may be due to the interaction with transport proteins that grant such localisation [87]. This interaction might be specific, as mentioned for so many other proteins within the N-terminal region and might be inhibited by the presence of exon 3 in some way. Nonetheless, it could also be the case that the interaction is electrostatic in nature and due to the presence of several acidic residues within exon 2 encoded sequence that would increase the negative charge of this particular area.

In line with its nuclear localisation, Tau protein has been demonstrated to bind to DNA in vitro [88,89] and in vivo [90] and RNA [91], as well as interact with chromatin components [92] and the inner side of the nuclear lamina [93]. Due to the specificity of isoforms found in nuclear Tau and the N-terminal region being the main interactor with other proteins and membranes, we cannot rule out that a portion of the N-terminal end encoded by exon 2 is responsible for these nuclear interactions of Tau.

Within the nucleus, Tau has been mainly reported to be found in the nucleolus and the pericentromeric heterochromatin [90,94], in both phosphorylated and dephosphorylated states, although the nucleolus exhibits primarily dephospho-Tau [54].

Tau functions in the nucleus have not been fully elucidated and are currently being intensively researched. Given the role of the nucleolus and its relationship to ribosomes and that Tau interacts with the ribosomal protein pS6 [95], Tau has been proposed to be linked to ribosome biosynthesis and regulation and ribosomal DNA transcription [68,95]. Nuclear Tau has also been proposed to participate in gene expression and DNA protection, for instance [54,96]; which could be linked to genome vulnerability and neurodegeneration found in tauopathies such as Alzheimer’s disease [54]. In fact, since Tau nuclear localisation occurs across different nervous and peripheral tissues, it has been proposed that it may actually carry out a more general role in genome surveillance [54]. 

As hinted before, all these nuclear functions of Tau, and more specifically of exon-2 containing isoforms of Tau, are susceptible of regulation through post-translational modifications, such as phosphorylation [97,98], further supporting the idea of post-translational modifications being a key factor of Tau’s functions in health and disease [58].

Finally, another interesting consequence of Tau’s position when the protein is attached to microtubules is that the projection domain regulates the spacing between microtubules in the axon and may therefore be at least partly responsible for axonal diameter [81]. The inclusion of exon 4a implies a great extension of the N-terminal region (Figure 4C), but the amino acid composition includes high proportions of proline, lysine, serine and glycine compared to other vertebrate globular proteins [63], much as the rest of Tau.

The primary repercussion of this elongation is an increased space between microtubules [99], which helps explaining the large diameter of the axon of peripheral neurons [81]. Such larger spacing may also contribute to axonal transport in these specific neuron populations with longer axons by reducing the resistance of the axoplasm, which would imply less energy is needed for such transport [56].

Apart from that, the possible function of exon 4a remains prominently understudied. Some authors theorise that Big Tau isoforms may be more related to axonal microtubule stabilisation than their lower-molecular-weight counterparts, which might be more associated to axonal growth instead [61], also exhibiting more dynamic neurites [56]. Nevertheless, given that the homologous to exon 4a in non-human primates [46] and non-mammals such as Xenopus [100] maintain the same size with small sequence correspondence, it has also been proposed that it might not be truly a functional region, beyond the enlargement of the projection domain [56,100].

### 4.2. Proline-Rich Region and Isoforms Including Exon 6

The proline-rich region constitutes a hinge between the N-terminal end and the microtubule-binding region and is characterised by an elevated proportion of proline residues (around 20% higher than the average for human proteins) [67], which contributes to an increased rigidity of this part of the molecule. In contrast with the amino-terminal area, this region is markedly basic and positively charged at physiological pH.

The PRR also exhibits a high content of serine and threonine residues, thus increasing its proneness to phosphorylation (with up to 22 predicted sites, 14 of which are serine) [67] and, in consequence, its susceptibility to drive Tau’s phosphorylated-mediated regulation [67,101]. Relatedly, the proline-rich area interacts with a great deal of kinases (such as fyn) and phosphatases (like PP2A/Bα); but also other signalling molecules, such as the isomerase Pin1, involved in Tau conformation regulation [67].

This region, however, is not just a regulatory or bridging area. Even though Tau’s function promoting microtubule assembly and stability has been typically linked to the appropriately-named microtubule-binding region, evidence show there is a less studied segment between residues K215 and N246 (and more specifically, the sequence 215KKVAVVR221) within the proline-rich region that also interacts directly with microtubules to exert such functions [102]. This role interacting with cytoskeletal proteins is further confirmed by studies showing that the PRR is involved in Tau’s association to actin [103]. Moreover, phosphorylation in the PRR directly affects its capacity to polymerise microtubules, further evidencing the importance of this role and the regulation of this region’s function in microtubule binding and assembly [101]. In fact, more recent evidence points that the PRR binds to tubulin in a strong, stoichiometric manner, while the microtubule-binding repeats have weaker but more distributed interacting sequences [81,104]. These authors thus proposed that the proline-rich region would constitute a “core tubulin-binding domain”, and the MTBD may participate increasing local tubulin concentration, hence facilitating polymerisation [104].

Other specific functions of this region are currently under research, including a role in Tau secretion [105] and participation in the interaction with PSD-95 in the postsynaptic area [106] and in the regulation of postsynaptic dysfunction via phosphorylation [107].

Inclusion of exon 6 implies the addition of another proline-rich sequence, right before the PRR (Figure 4C), which can be interpreted as an extension of this region, increasing the rigidity of the molecule and constituting another area susceptible to phosphorylation and proteolysis that can help regulate Tau’s functions [9,45,65]. 

Isoforms containing the canonical version of exon 6 are found in higher amounts in adult spinal cord, peripheral nervous system and skeletal muscle [65], partly coinciding with the expression pattern of isoforms containing exon 4a [56]. In fact, the longest Tau isoform described results from the inclusion of both exons in skeletal muscle [9] (Figure 2A). The larger, more rigid bridging region generated upon exon 6 inclusion may also determine differences in microtubule spacing [65], which may be related to this expression pattern in regions where microtubule spacing is increased [56,64,65]. 

Analysing the temporal and spatial distribution of isoforms including exon 6, some authors have proposed that they can be found in developmental stages, subcellular locations and tissues where a more dynamic cytoskeleton might be needed and might then be linked to neuronal plasticity and axonal guidance functions [9,65]. However, these isoforms have proven to inhibit neurite extension in SH-SY5Y cells, which suggests a regulatory role more than an active one in these functions [65].

Beyond that, exon 6 has a highly conserved sequence, suggesting the interaction with specific molecules, as occurs with exons 2 and 3 [9], although these ligands have not been clearly established yet [9,65]. 

On the other hand, when either the proximal or distal alternative 3′ splice sites are used, the corresponding isoforms 6p and 6d are generated [64]. Both of them find a premature stop codon (Figure 3A), generating truncated isoforms that do not include a proline-rich region at all, nor do they present the MTBD or the C-terminal end [9,45,64] (Figure 4D). Thus, these isoforms are not expected to be able to carry out any of the functions associated with this areas such as microtubule binding [65] and may see the functions associated to the N-terminal end altered, either being enhanced due to an increased availability of NTR residues or being hindered due to the lack of other regions that regulate NTR-related functions, as seems to be the case for kinesin-based axonal transport that are inhibited by 6p and 6d isoforms [45].

Very relevantly, LaPointe et al. proved that both 6p and 6d isoforms are able to inhibit in vitro polymerisation of other, full-length Tau isoforms, with 6p isoforms constituting more potent inhibitors than 6d ones, although the presence of the N-terminal inserts encoded by exons 2 and 3 increases the potency of the latter ones [45]. 6p and 6d isoforms reduce the number of filaments and the overall mass, while remaining soluble, so they proposed a model in which they stabilise a conformation of Tau that is not aggregation-prone, very much as other N-terminal fragments previously tested by the same group [108].

It remains unclear whether this inhibition might occur in vivo as well, since 6p and 6d isoforms are in comparably very low amounts in the human brain, with the exception of the cerebellum [9,45,64]. Nonetheless, the punctuated immunoreactivity of an specific antibody against 6d isoforms may indicate local enriched levels of these isoforms in specific subcellular locations [45]. The authors also point out that, although the in vitro experiments required higher amounts of 6p and 6d isoforms, these had to compete with aggregation inducers, so the required levels may not be as high in a cellular context. On the other hand, given the higher expression of these isoforms in cerebellum and the diminished vulnerability of this area to tau lesions in AD, it is tempting to establish a role for such isoforms as aggregation inhibitors in vivo [45].

Lastly, both 6p and 6d isoforms are generated due to a shift on the reading frame after the use of the alternative sites and hence include two unique sequences (PCCVPRATFLS and FWSKGDETQGG, respectively) (Figure 3A), which may possess specific functions that remain unexplored [45].

### 4.3. Microtubule-Binding Domain: A Repetitive Region

Tau is a member of the microtubule-associated proteins (MAP) family and as such, microtubules are the main ligand of the protein [3,67]. This interaction is carried out between microtubular tubulin and Tau’s proline-rich region and microtubule-binding tandem repeats (MBTR) [67,102,104]. 

These microtubule-binding tandem repeats are encoded by exons 9–12, which implies that the inclusion or exclusion of exon 10 determines differences in length of this region [9,32] (Figure 4B). Namely, each repeat is composed of a highly conserved 18 amino acid sequence and 13- or 14-residue separating sequences [67,81]. The inclusion of exon 10 entails the increase from 3 to 4 tandem repeats, which directly affects functionality, with 4R isoforms displaying a greater ability to bind to microtubules than their 3R counterparts [3,32,81]. Apparently, the main driver of this difference of affinity between 3R and 4R isoforms is the region between the first and the second repeat, through the sequence 275KVQIINKK282 [81] (number according to Tau 441). The extension of this region may also have other consequences, such as alterations of Tau’s subcellular location mediated by non-specific, sized-determined sorting mechanisms [80], that might operate in the same way for other extensions of the molecule such as exon 4a or exon 6 inclusion (Figure 4B).

The MBTR are very similar to the PRR in terms of charge and isoelectric point, being quite basic and positively charged, which facilitates interaction through electrostatic interactions with negatively charged glutamate-rich tubulin C-termini [67], which constitute a rather exposed binding site and is thus consistent with the extremely dynamic “kiss and hop” interaction between Tau and microtubules [67,109].

Besides the MBTR and the PRR interaction with microtubules, a pseudorepeat region has been described within the C-terminal side, which is also present in other members of the MAP family [110]. 

As the rest of the mentioned functions for the protein, microtubule binding is modulated by phosphorylation. S262 phosphorylation is a potent inhibitor of microtubule binding within the MBTR, but the majority of phosphorylation sites affecting microtubule binding are located either in the PRR or the CTR [101,102,111]. Interestingly, microtubules compete with phosphatase PP2A for binding to residues 224 to 236 within the MBTR, conferring the MTBR a role as an indirect modulator of Tau’s phosphorylation [81]. 

Microtubules participate on a myriad of cellular processes, including cell morphogenesis and division and intracellular trafficking of organelles, lysosomes, endocytic and exocytic vesicles, etc. [67,81]. Consequently, Tau would act as a modulator of all these functions via interaction with microtubules, which explains Tau’s influence on axonal transport and pose an important regulatory role of Tau phosphorylation with cellular-wide consequences. 

In this respect, other post-translational modifications have been proven to modulate Tau’s functions [58]. More specifically, acetylation is crucial to the microtubule-binding function, since it neutralises Tau’s charge and thus hinders this binding, especially in the MBTR, where lysine residues are overrepresented [67]. This is notably relevant, given that Tau itself has been demonstrated to exert acetyltransferase activity and could then self-regulate in this way [67,112]. 

Beyond binding to microtubules, the MBTR region has been proven to interact with many other proteins, including actin, the low-density lipoprotein receptor-related protein 1 (LRP1) or heat-shock proteins, among others [67,113]. Functionality-wise, these proteins are related in its majority with microtubule-related processes and signalling mechanisms, but also some of them with cell death mechanisms [67].

Of particular interest is that Tau uses this region to bind to itself [113,114] and can be related to self-aggregation and polymerisation into filaments and neurofibrillary tangles, typical of tauopathies such as Alzheimer’s disease [48,76,81,114,115,116]. This may help explain the fact that isoforms lacking this region such as 6p and 6d isoforms do not aggregate and even possess anti-aggregative properties [45]. In line with this, there are sequences from the second and the third repetition displaying β-structures (an uncommon feature on an intrinsically disordered protein as Tau), which can assemble between themselves and with other such structures from Tau molecules in the vicinity [67].

Together, these data underlined the importance of the microtubule-binding region on physiological and pathological Tau functions, and stresses its role in tauopathies such as AD, where certain parts of this region—and of the C-terminal end—are found at the core of filaments that polymerise giving rise to such lesions [117,118]. 

### 4.4. W-Tau: The Uncharted Territory of Novel Isoforms and New Mechanisms

The discovery of novel, human-specific Tau isoforms generated by intron 12 retention implies the opening of a new avenue of research that has never been explored, both in terms of isoform properties and splicing mechanisms of the *MAPT* gene [37]. 

W-Tau isoforms are generated from the retention and partial translation of intron 12, that results in the appearance of a premature stop codon followed by a canonical poly-A signal, truncating the C-terminal end (Figure 4D) and causing the translation of a unique 18-amino-acid sequence containing two tryptophan (W) residues (KKVKGVGWVG CCPWVYGH) [37]. 

The resulting isoforms retain the rest of the molecule up to the last microtubule-binding repeat, and can potentially harbour, then, zero, one or two N-terminal inserts and three or four microtubule-binding repeats. This would make W-Tau isoforms strikingly similar to a fragment described by Zhang et al. [119], generated by asparagine endopeptidase cleavage that comprises residues 1–368 (numbered according to the sequence of Tau 441). Indeed, W-T42 (4R2N) would share the exact same sequence with the exception that asparagine endopeptidase truncated Tau would end in N368 and W-T42 would be followed by the specific 18-residue sequence [37] (Figure 3B).

Notably, Tau 1–368 proved to be prone to aggregation and a strong inducer of neurodegeneration, while being unable to assembly microtubules and promote axon elongation [119]. Indeed, Tau truncation has been proposed to be one of the post-translational modification most intimately linked to Alzheimer’s disease onset and progression from early stages, since it induces Tau misfolding and self-aggregation, as well as neurodegeneration; all correlating to cognitive decline [83]. However, contrary to what one would expect and in spite of the sequence similarity, W-Tau isoforms have been demonstrated to keep the ability to bind to microtubules and exhibit an increased solubility and a non-aggregative behaviour compared to both full-length Tau and Tau 1–368 [37]. In this sense, and in their lower expression in the brain with respect to the six main isoforms, W-Tau intron 12 retention-mediated truncation is more similar to that of isoforms 6p and 6d [45] than to a post-translational truncation; although the ability of W-Tau to influence the aggregation of other isoforms as 6p and 6d isoforms do is still to be determined.

The reasons behind the different behaviour observed for different truncated isoforms remains unclear and purely speculative for the time being. On the one hand, it appears that those truncated isoforms that are generated by means of alternative splicing—that is, W-Tau and 6p and 6d isoforms—have an inhibitory effect on Tau aggregation [37,45], while truncation as a post-translational modifications clearly promotes it [83,119]; which may point towards a regulating role for alternative spliced variants whereas those generated from proteolysis might be related to pathological situations [83]. 

On the other hand, Fasulo et al. proved by deletion mapping experiments that the shortest fragment with toxic, apoptotic properties is the peptide constituted from residues 151–368 of Tau 441 [120]. This may explain the difference between W-Tau isoforms and their asparagine endopeptidase-mediated equivalents, with the 18-residue sequence following N368 either masking or blocking the interaction with this specific residue or altering the conformation of the whole molecule, thus hindering the exertion of aggregation-promoting activities.

The mechanisms that explain the non-aggregative properties of these isoforms also are pending to be studied [37]. The authors point out that these properties may be explained in part by the 18-amino-acid sequence specific to W-Tau isoforms. Namely, they propose that the sequence GVGWVG is similar in nature to that of some recently described inhibitors of Tau and amyloid β aggregation [121]. Also, this sequence contains two adjacent cysteines, a feature that has previously been reported to be able to produce an intramolecular ring [122] that can alter Tau’s conformation and determine a change on its properties. In addition, the truncation of the C-terminal region implies the loss of a small sequence within exon 13 that can be found in the core of Tau filaments, as previously mentioned [117,118], maybe hindering filament-seeding to some extent.

No studies have been done so far regarding W-Tau subcellular localisation. It is expected that W-Tau isoforms with 0N, 1N and 2N would distribute accordingly throughout the cell [53], but there is an interesting phenomenon to be taken into account: the start of the 18-residue sequence is fairly similar to that of certain nuclear localisation signals [123,124], so a specific role for W-Tau isoforms determined by a different subcellular localisation cannot be ruled out.

W-Tau isoforms were found both in non-demented human and in Alzheimer’s patients’ hippocampus and frontal lateral cortex as mature mRNA species and as protein, using a specific antibody generated against the 18-residue sequence arising from intron 12 retention. Noteworthily, the levels of W-Tau in Alzheimer’s brains were diminished with respect to non-demented subjects, more prominently so in advanced stages of the disease whereas total Tau displayed an accumulating pattern [37]. Given this evidence suggesting an inverse relationship with Alzheimer’s disease and the analogy between W-Tau and 6p/6d containing isoforms, future studies would benefit from exploring the expression of such isoforms in other regions that are less vulnerable to tau pathology, like the cerebellum, where 6p/6d isoforms are found in greater amounts [45].

As for the mechanism that gives rise to this novel set of isoforms, it has been proposed that *MAPT* intronic sequences might have a regulatory role in the development of filamentous inclusions typical of tauopathies, since tau lesions are not reproduced in other species, not even great apes with highly similar (if not identical) Tau protein amino acid sequence [46]. In this sense, it is important to underline that W-Tau isoforms would be human-specific and that animal models poorly reproduce Alzheimer’s disease multifactorial pathology [76]. All this, together with the diminished levels of W-Tau in Alzheimer’s disease patients, might suggest that intronic sequences could have a modulating effect by means of generating species-specific isoforms through determined alternative splicing mechanisms such as intron retention. Such mechanism would imply the need of finely regulated splicing mechanisms dealing with intronic sequences; an idea that can be supported by the fact that the protein saitohin is encoded within an intronic sequence of the *MAPT* gene [38] and that exon 4a might actually come evolutionary from an intron of other gene [56]. 

Such evidence, together with W-Tau’s modulation in Alzheimer’s disease patients, suggest it could be interesting to explore these isoforms and the mechanisms leading to their generation from a clinical perspective. Therapeutic correction of aberrant splicing [125] and modulation of both intron retention [126] and post-translational modifications such as truncation [58] have been previously proposed as potential therapeutic aims. In this line, an increase of this non-aggregative species or the prevention of its decline in Alzheimer’s disease might turn out to be a valuable strategy in the future [37].

### 4.5. The Interplay: Interaction between Functional Regions

We cannot finish this review without highlighting that the functional regions that compose Tau protein are not isolated one from the other but together as a whole forming the protein, and most of their functions cannot be pinpointed to a specific site, but rather, to the interplay of more than one of these functional regions. 

For instance, Tau is an example of an intrinsically disordered protein, characterised by low sequence and structural complexity [67], which implies that Tau conformation cannot be described as a single state, but rather a conformational ensemble, greatly depending on Tau’s post-translational modifications, as well as the specific sequence of a given isoform, its environment, binding proteins, etc [127]. Incidentally, it is important to consider that not the whole protein is equally disordered, but the NTR and the PRR show the highest degree of disorder, partly explaining the great extent of binding promiscuity of these regions [67].

Among these Tau conformational states, the paperclip conformation has been greatly studied and proposed to be a usual conformation of soluble Tau, at least in vitro. Such conformation requires the physical approach of the N-terminus and the C-terminus of the Tau molecule [128]. It is not surprising, then, that modifications in the C-terminus, such as the pathological mutation R406W or pseudophosphorylation of certain sites have a direct effect on NTR-mediated interactions [67,81]. The paperclip conformation is proposed to be crucial for Tau’s physiological functions and can be greatly disturbed by modifications such as phosphorylation or truncation on either end [127,128]. 

In fact, this conformation can inhibit self-aggregation by masking of the sticky domains of the MBTR [67], which may contribute to understanding the increased tendency towards aggregation of truncated isoforms in either the N- or the C-termini [83]. Isoforms 6p and 6d do not have that problem despite truncation because they lack the microtubule-binding repeats themselves [45] and, as for W-Tau isoforms, it has proposed that the 18-residue sequence they present after the repetitions might trigger a different conformation that elicits a similar result [37]. 

However, the importance the interaction between different parts of the protein applies to other functional regions of Tau as well. For example, it has been consistently proven that for microtubule binding and assembly functions of Tau to be efficient, intramolecular interactions between the PRR and the MTBR are needed [101,102]. In fact, such interactions are susceptible of modulation, since they can be reduced by the NTR through, precisely, a conserved conformational ensemble; negatively regulating tubulin binding to both the PRR and the MTBR [104].

The regulation between different areas is not unidirectional either. Phosphorylation at some residues of the PRR can deeply alter interactions from the N-terminal end and the MTBD. Actually, the majority of the regulatory sites of MBTR-mediated aggregation are present in either the PRR or the CTR; such as interaction with microtubules and polymerisation being inhibited by phosphorylation of serine 214 [67]. The regulation of these sites is intimately associated to AD onset and progression, with the AT8 site (S202/T205) being the best correlated phosphorylation with disease progression, but also including other important ones as S199 or T231 [8,129,130,131,132,133]. Phosphorylation on these sites hinders Tau’s microtubule-related functions (even if it is not sufficient to abolish microtubule binding and polymerisation) and might have modulating effects on other Tau functions such as intracellular signalling [67].

Collectively, these results make clear that Tau protein is much more than a microtubule-associated protein and that their functions are deeply regulated by a great deal of external factors including alternative splicing and post-translational modifications, but also the complex interplay between the different functional regions of the protein.

## 5. Conclusions

In the past few decades, our understanding of the complex mechanisms that reign and modulate alternative splicing has increased greatly, as have our knowledge of the implication that these processes might have in physiological and pathological conditions [12,17,28,29,31,34].

Specifically, Alzheimer’s disease and other age-related pathologies have been proposed to be, at least partly, consequence of aberrant or unsuccessful alternative splicing events [17,49]. Even so, some evidence point out that alternative splicing may not be, after all, the great driver of proteomic diversity that it was originally thought to be [14], potentially leading to underestimate the influence it might have in physiological and pathological processes.

Nevertheless, it seems very clear that the *MAPT* gene is genuinely subjected to alternative splicing, with current estimates calculating over 50 isoforms can be generated by these mechanisms [9,14,15,37,45]. This paints a clear picture of the direct influence the alternative splicing process might have in the onset and development of Alzheimer’s disease and other tauopathies [17,49].

Considering the huge variety of alternatively spliced Tau isoforms and that some of them have just been discovered [37], there is a deeply underresearched niche, including the functions and distribution of isoforms that express exon 4a, exon 6 in either of its alternative splice sites or retain intron 12, as well as the possibility of new isoforms arising from other, less studied alternative splicing processes, such as other intronic-related events. Charting these previously unexplored research avenues surely offers an exciting field that we will most likely see thrive in the upcoming years.

## Figures and Tables

**Figure 1 cells-11-00840-f001:**
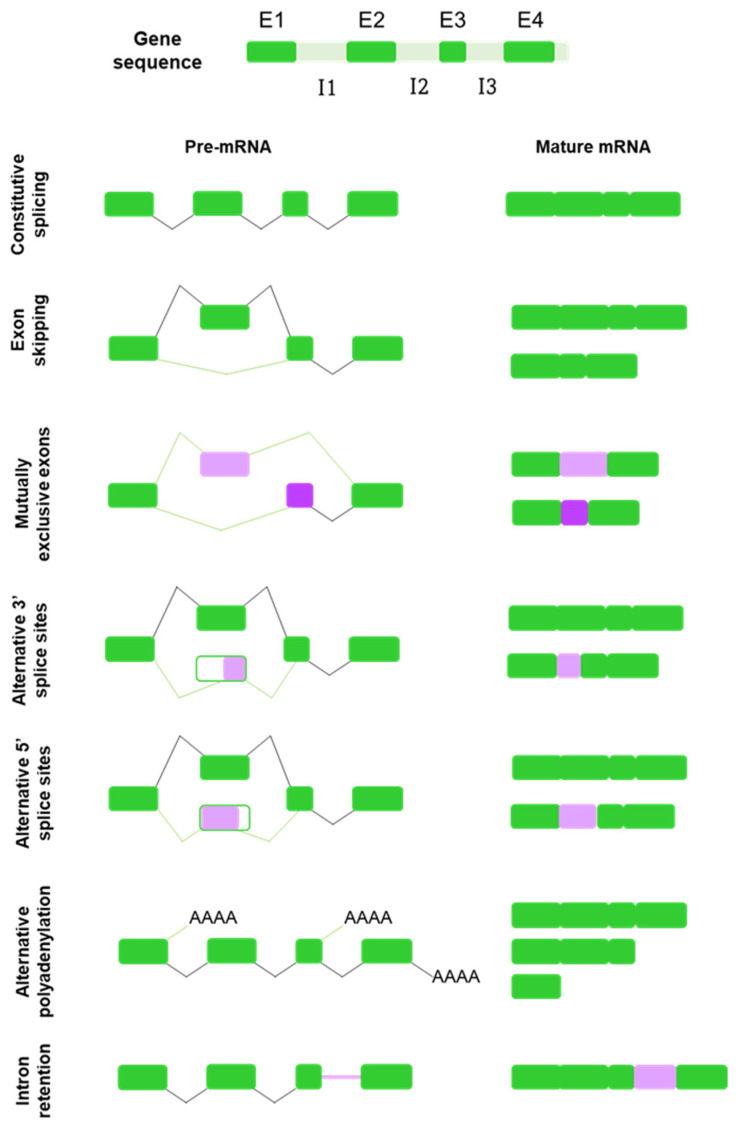
Schematic representation and summary of the mechanisms of alternative splicing. Pre-mRNA from a hypothetical gene is displayed with 4 exons (E1–E4, green) and 3 introns (I1–I3, light orange). Below, schematic representations of the different mechanisms of alternative splicing is evidenced as different potential alternative splicing decisions marked with purple branches as opposed to black branches for constitutive splicing decisions. RNA and protein fragments marked in violet and purple (mutually exclusive exons and alternative 3′ and 5′ splicing sites) show the differences with respect to constitutive splicing. “AAAA” (alternative polyadenylation) represents a polyadenylated sequence. RNA and protein fragments marked in light orange (intron retention) point out intronic regions that are maintained instead of spliced out, while the black dashed line represents the constitutive splicing decision that would happen if intron retention did not take place.

**Figure 2 cells-11-00840-f002:**
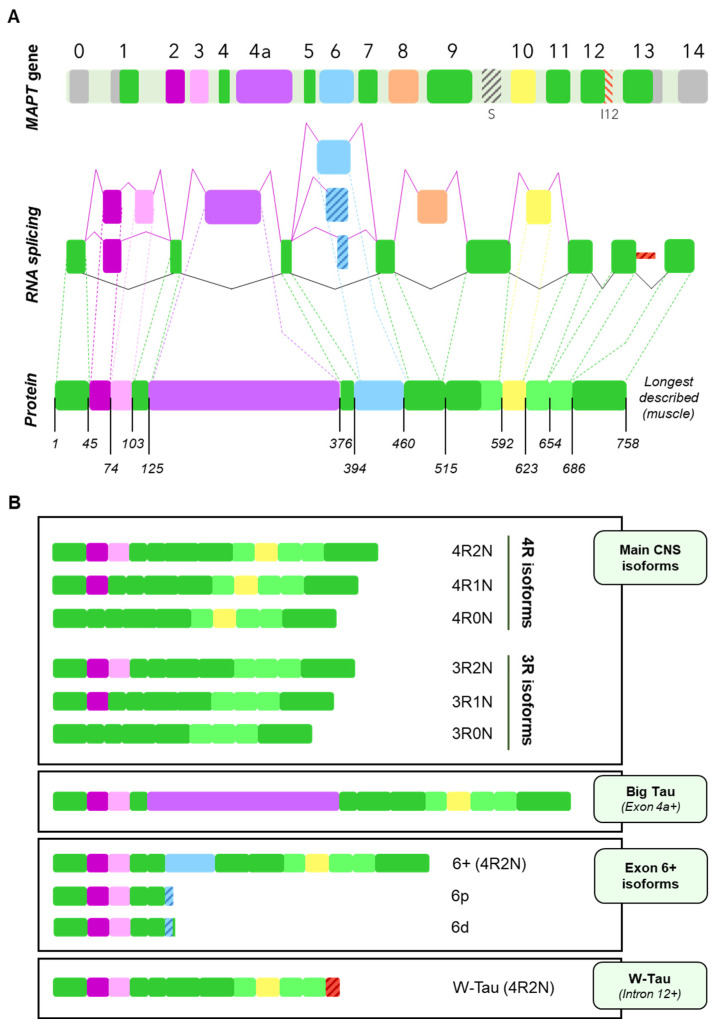
Alternative splicing of the *MAPT* gene. (**A**) Schematic representation of the splicing process. The *MAPT* gene is shown with its 16 exons highlighted in different colours. Exon 0 and part of exon 1 comprise the 3′ untranslated region while the end of exon 13 and exon 14 make up the 5′ untranslated region and are all marked in grey. Constitutive exons (exons 1, 4, 5, 7, 9, 11, 12 and part of exon 13) are displayed in green. Exons 2, 3, 4a, 6, 8 and 10 are subjected to alternative splicing and have their own colours (purple, pink, violet, blue, orange and yellow, respectively). The dark-grey stripped fragment in the intron between exons 9 and 10 (S.) represents the nested gene encoding the protein saitohin. The red stripped region in the intronic area between exons 12 and 13 (I12) represents the part of intron 12 that is retained in W-Tau isoforms of Tau. The colour patterns are maintained in the RNA splicing schematic representation, stripped boxes symbolising splicing decisions that would be responsible for truncated isoforms of Tau. Finally, the longest isoform of Tau described is depicted below, including all the constitutive exons and exons 2, 3, 4a, 6 and 10 from those subjected to alternative splicing. (**B**) Representation of the main type of isoforms of Tau that can arise from the alternative splicing of the exons depicted above. The six main isoforms found in the Central Nervous System are displayed on the first box, including 4R and 3R isoforms with 0, 1 or 2 N-terminal inserts. For isoforms including exons 4a (Big Tau) or 6 or retaining intron 12 (W-Tau), only the 4R2N isoforms are depicted, but note that all combinations are potentially possible.

**Figure 3 cells-11-00840-f003:**
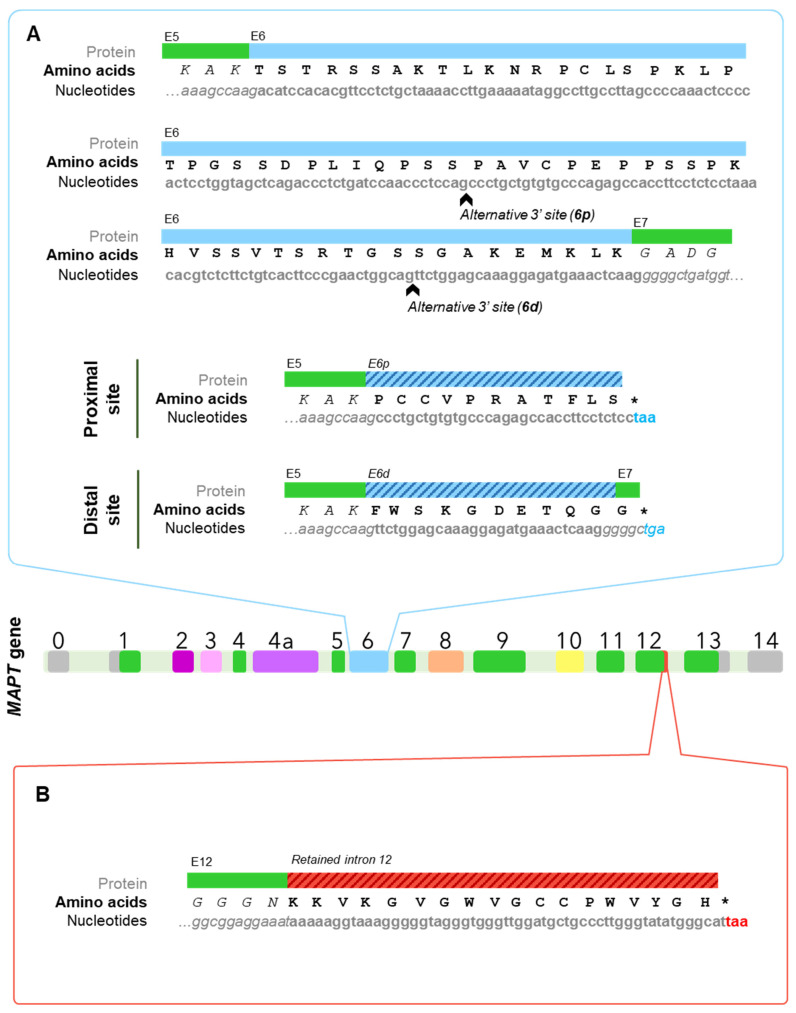
Translation of truncated Tau isoforms lacking the C-terminus. (**A**) Nucleotide and amino acid sequence of exon 6, flanked by exons 5 and 7. The alternative 3′ splice sites of exon 6 that generate a shift of the reading frame are indicated with black arrows within the exon sequence. The sequence resulting from such frameshifts in the proximal and distal sites are specified below. (**B**) Nucleotide sequence of the end of exon 12 and the beginning of intron 12 and the amino acid sequence that would be translated into upon intron 12 retention, giving rise to truncated W-Tau isoforms.

**Figure 4 cells-11-00840-f004:**
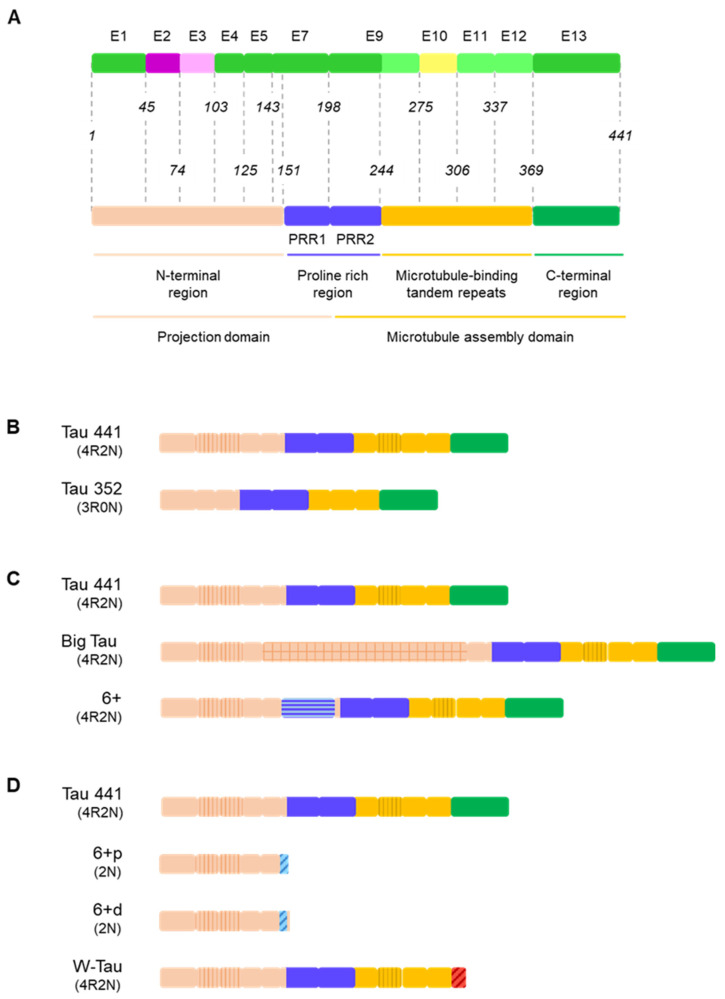
Functional regions of Tau protein. (**A**) Schematic representation of the equivalence between Tau amino acids translated from each exon and regional functions of Tau protein. (**B**) Differences on the length of different functional regions due to the inclusion or exclusion of exons 2, 3 and 10, indicated by vertical stripes. Tau 441 (4R2N) and Tau 352 (3R0N) are shown to highlight the differences. (**C**) Differences on the length of different functional regions of Tau due to the expansion of the protein with respect to Tau 441 (4R2N). Inclusion of exon 4a (pink, squared area) implies the extension of the N-terminal region, while inclusion of exon 6 (blue, horizontally stripped area) extends the molecule including a proline-rich exon that would extend the proline-rich region. (**D**) Differences on the length of Tau functional regions in isoforms lacking the C-terminal region. 6p and 6d isoforms lack the proline-rich region, the microtubule-binding region and the C-terminal end altogether, while W-Tau isoforms only lose the C-terminal end. Blue, diagonally stripped regions on isoforms 6p and 6d represent the translation of their respective specific sequences, which can have their own functions. Red stripped regions on the W-Tau isoform represent the unique 18 amino acid sequence characteristic of these isoforms, which may also have specific properties.

## Data Availability

Not applicable.

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
