# Peer review of "What’s in a Gene? The Outstanding Diversity of MAPT"

_cells, 2022, doi:10.3390/cells11050840_

Round 1

Reviewer 1 Report

The authors reviewed the literature on the structure of the MAPT gene and the generation of several transcripts by alternative splicing, as well as the role of different tau protein isoforms in normal physiological conditions and in various diseases called tauopathies, including Alzheimer's disease.

The review is well written, but despite detailing all the different MAPT isoforms and their role in different human tissues, it completely lacks a description of tau isoforms observed within the cell nucleus and nucleolus and its possible interaction with DNA, or with RNA. I believe that a part relating to the different isoforms of the MAPT gene connected with the function of the nuclear tau is indispensable.

Moreover, there are some typos and some inaccuracies to correct:

The sentence in the abstract “Tau protein has been proven to be subjected to alternative splicing…” (line 18) should be corrected, considering that it is not the tau protein that undergoes alternative splicing, but the MAPT gene.

References [3] and [4] are the same article. One of them should be eliminated.

Figure 1: The last column is marked as "Protein", but I think it is better to call this column "mature mRNA"

In line 200: "N0, N1, and N2 isoforms ..." should be rewritten as "0N, 1N, and 2N isoforms ..."

The word "Tay" at the end of line 388 should be correct in "Tau".

Figure 4: In sections B, C, and D, the name of 4R2N tau isoform should be uniformly indicated in the same way.

Line 765: the references [9, and 115] are probably not the best ones to mention the role of AT8.

Author Response

Dear Reviewer, 

Thank you kindly for your comments and suggestions, we think your contibution truly enriched our review, specially with the idea of including some information on nuclear functions of Tau. 

Please, find below our point-by-point response to your comments:

Comments and Suggestions for Authors

- The authors reviewed the literature on the structure of the MAPT gene and the generation of several transcripts by alternative splicing, as well as the role of different tau protein isoforms in normal physiological conditions and in various diseases called tauopathies, including Alzheimer's disease.

The review is well written, but despite detailing all the different MAPT isoforms and their role in different human tissues, it completely lacks a description of tau isoforms observed within the cell nucleus and nucleolus and its possible interaction with DNA, or with RNA. I believe that a part relating to the different isoforms of the MAPT gene connected with the function of the nuclear tau is indispensable.

The authors thank the reviewer for this valuable insight and for pointing out this fundamental part of the scientific literature that we firstly overlooked. We have included it in several points throughout the text and have discussed it accordingly. Namely, we have included:

  • A brief, introductory paragraph to the matter of different subcellular localisations for the different isoforms (lines 255-268).
  • A more detailed discussion of isoforms that are located in the nucleus and nucleolus, within the N-terminal region functions section (lines 487-523).
  • A brief speculative note on the possible nuclear localisation of W-Tau isoforms, in the W-Tau functions section (lines 741-746).

There is a vast amount of information regarding Tau’s functions within the nucleus and nucleolus in the scientific literature, but sadly a rather big part of those studies lack information regarding Tau isoforms involved, which frankly is a widespread problem within different fields. Thus, we believe this may be enough to cover what we know for certain of the involvement of different Tau isoforms in nuclear functions, without dwelling too deep in such specific functions, as it was not the intention of this review.

Moreover, there are some typos and some inaccuracies to correct:

The sentence in the abstract “Tau protein has been proven to be subjected to alternative splicing…” (line 18) should be corrected, considering that it is not the tau protein that undergoes alternative splicing, but the MAPT gene.

We wish to thank the reviewer for pointing this out. It has been changed to “MAPT transcripts have been proven to be subjected to alternative splicing”.

References [3] and [4] are the same article. One of them should be eliminated.

We thank the reviewer for noticing this too, reference number 4 has been eliminated and the corresponding numbering modified accordingly.

Figure 1: The last column is marked as "Protein", but I think it is better to call this column "mature mRNA"

We agree with the reviewer that protein is maybe not the best label for the products shown in the figure and have consequently changed it, as they suggested.

In line 200: "N0, N1, and N2 isoforms ..." should be rewritten as "0N, 1N, and 2N isoforms ..."

This has been modified accordingly.

The word "Tay" at the end of line 388 should be correct in "Tau".

It has been corrected as the reviewer pointed out.

Figure 4: In sections B, C, and D, the name of 4R2N tau isoform should be uniformly indicated in the same way.

It has been modified, so that all of them display “Tau 441 (4R2N)”.

Line 765: the references [9, and 115] are probably not the best ones to mention the role of AT8.

New references have been added to this point, including more classical ones referring to AT8 (now in line 820).

Reviewer 2 Report

Dear authors,

the review is very well done. It describes tau protein, its structure, many isoforms derived from alternative splicing, their expression in human and non-human and their possible implication in the development of tauopathies such as Alzheimer's disease. The manuscript is clear and easy to read and the figures are well done. I advise the publication of this manuscript. I think that this review is very well written. So I accept it in the present form.

Author Response

Dear reviewer, 

Please, find below our response to your comment. We want to thank you for being so kind and accepting our manuscript. You will find the manuscript improved upon the inclusion of the other reviewers' suggestions too, so we hope you would still consider it suitable for publication. 

Dear authors,

the review is very well done. It describes tau protein, its structure, many isoforms derived from alternative splicing, their expression in human and non-human and their possible implication in the development of tauopathies such as Alzheimer's disease. The manuscript is clear and easy to read and the figures are well done. I advise the publication of this manuscript. I think that this review is very well written. So I accept it in the present form.

We sincerely thank the reviewer for their kind words and encouragement.

Reviewer 3 Report

The review by Ruiz-Gabarre et al. is focused on the MAPT gene, with main interest in its several different isoforms obtained by alternative splicing.

It is a very detailed review, with a series of interesting and precise information. Even the figures are clear and useful to better understand the complex pictures of the different isoforms. I have no major objection, just some comments.

Main comments

1- In the description of the different isoforms, it is not always clear if the reported findings or supporting evidence derived from transcript (e.g. RNAseq) or protein analysis.

2-“W-Tau has been discovered to be expressed in ~50-75% of humans” (line 348)

How much is reliable this finding? It is possible to firmly state that there are “normal” subjects with and without this isoform?

3- It would be noteworthy if the authors can directly link some of the data about the isoforms, their expression and roles… with known mutations in MAPT and associated phenotypes. Only in a couple of sentences (line 439, line 740) there is such information, despite the plenty of details about different isoforms.

Minor changes

-“Tau protein has been proven to be subjected to alternative splicing” (abstract)

The transcript or mRNA is subjected to splicing, not the protein

-lines 297-298: if derived from the same gene, it is difficult to not consider it as a MAPT isoform (but maybe here the statement is provocative)

Author Response

Dear reviewer, 

We want to kindly thank you for all your comments, we think your contribution helped us to improve this manuscript, specially the comment regarding the possible confusion between protein and mRNA results, which we wanted to be very clear and easily identifiable. 

As for the rest, please, find below our point-by-point response to your comments and suggestions:

The review by Ruiz-Gabarre et al. is focused on the MAPT gene, with main interest in its several different isoforms obtained by alternative splicing.

It is a very detailed review, with a series of interesting and precise information. Even the figures are clear and useful to better understand the complex pictures of the different isoforms. I have no major objection, just some comments.

Main comments

1- In the description of the different isoforms, it is not always clear if the reported findings or supporting evidence derived from transcript (e.g. RNAseq) or protein .

We would like to thank the reviewer for their comment, since we intend to be as transparent as possible with this to generate a solid review that explains the origin of each finding.

As a general statement, most of the findings have been explained in a way that, when referring to protein analysis results, the terms “protein” or “isoform” are employed, while when referring to RNA there is a explicit mention to this or the technique used (RT-qPCR, RNAseq).

However, the reviewer is right and, upon revision, some points could have been further clarified. Thus, we have included some references throughout the text to underline this point:

  • “as protein”, line 192.
  • “of these proteins”, line 234.
  • “Higher molecular weight Tau protein”, line 281-282.
  • “as a protein”, line 299.
  • “RNAseq data pointed out that mRNA for W-Tau is expressed in ~50-75% of humans”, line 364
  • “mRNA and protein”, line 370.

As for the second section of the review where we focus on the functions of different Tau isoforms, we believe it is clear from the beginning and the schematic representation on Figure 4 that we refer to protein, but again, several remarks can be found after each assertion, so we do not think we would need to add any further references.

2-“W-Tau has been discovered to be expressed in ~50-75% of humans” (line 348)

How much is reliable this finding? It is possible to firmly state that there are “normal” subjects with and without this isoform?

The percentage of TIR-MAPT expression in human brain was retrieved from 363 samples of 3 brain regions (frontal cortex,  dorsolateral prefrontal cortex and hippocampus) of 180 human brain healthy donors were retrieved from the Genotype-Tissue Expression GTEx dataset,  in which next-generation RNA sequencing using Illumina HiSeq 2000 was performed (Lonsdale, Thomas et al. 2013). In that study, RNA sequencing (RNA-seq) uses a 76-base, paired-end Illumina TruSeq RNA protocol, averaging ∼50 million aligned reads per sample to maximize sequencing value. That makes possible to accurately measure moderately expressed transcripts, as well as some with low-level expression. In such conditions, in 25-50% of the brain samples intron 12 retaining MAPT reads (TIR-MAPT) are not detected (García-Escudero, Ruiz-Gabarre et al. 2021), however, it cannot be ruled out TIR-MAPT expression under the detection threshold. In fact, at protein level assayed by Western blot, all brain samples show detectable W-Tau levels although the number of samples was smaller (n=42).

We have now clarified that this percentage was referred to mRNA, not to the protein levels (line 364) and we have added “or that the expression is so low that it falls below the technique’s detection threshold”, line 369.

García-Escudero, V., et al. (2021). "A new non-aggregative splicing isoform of human Tau is decreased in Alzheimer’s disease." Acta neuropathologica 142(1): 159-177.         

Lonsdale, J., et al. (2013). "The genotype-tissue expression (GTEx) project." Nature genetics 45(6): 580-585.

3- It would be noteworthy if the authors can directly link some of the data about the isoforms, their expression and roles… with known mutations in MAPT and associated phenotypes. Only in a couple of sentences (line 439, line 740) there is such information, despite the plenty of details about different isoforms.

We wish to thank the reviewer for this great suggestion. We agree that, indeed, mutations in MAPT and their consequent modification of Tau sequence can have a great relevance and influence on Tau’s functions, both in pathological and physiological conditions. However, the aim of the second part of this review was to highlight the role of the different regions of the protein and the protein as a whole in basal, physiological conditions. In the text we only mentioned these mutations as a reinforcement of what was being explained (line 439) and an example of something that can modify Tau’s conformation (line 740) but did not dwell on any of them any further. Thus, we feel this specific topic falls beyond the scope of our review. Additionally, due to the great volume of scientific literature that exists regarding Tau mutations and their potential implication in Tau’s functions, we think it would greatly extend the present work, if included.

Minor changes

-“Tau protein has been proven to be subjected to alternative splicing” (abstract)

The transcript or mRNA is subjected to splicing, not the protein

This point was also noted by another reviewer. We want to thank them both and have accordingly corrected it to “MAPT transcripts have been proven to be subjected to alternative splicing”.

-lines 297-298: if derived from the same gene, it is difficult to not consider it as a MAPT isoform (but maybe here the statement is provocative)

We wish to thank the reviewer for their insightful remark on this topic. However, whether coming from a single gene is enough to be considered the same protein is somewhat doubtful to us, given the very example of the MAPT gene that encodes the protein saitohin, a completely different protein from Tau, within one of its introns.

Nevertheless, the comment on lines 297-298 was indeed provocative, intended to introduce the reasoning below of Tau being much more than a microtubule-associated protein. To avoid any confusion we have slightly changed it to “Importantly, these would be the only Tau isoforms to lack the microtubule-binding domain, which begs the question as to whether they can be considered Tau isoforms at all if we were to focus solely on its function, given that the very core function of Tau as a member of the microtubule-associated proteins, would be related to microtubules“.

Round 2

Reviewer 1 Report

The authors updated and corrected the manuscript following all the suggestions indicated. In my opinion, the present version of the review can be published on Cells.